# Predictive value of somatic and functional variables for cognitive deterioration for early-stage patients with Alzheimer's Disease: Evidence from a prospective registry on dementia

Liane Kaufmann[1]*, Tilman Gruenbaum[1,2☯], Roman Janssen[2,3☯], Elisabeth M. Weiss[2], Thomas Benke[4], Peter Dal-Bianco[5], Michaela Defrancesco[6], Gerhard Ransmayr[7], Reinhold Schmidt[8], Elisabeth Stögmann[9], Josef Marksteiner[10]*

1 Department of Neurology and Clinical Neuropsychology, Ernst von Bergmann Klinikum, Potsdam, Germany, 2 Institute of Psychology, University of Innsbruck, Innsbruck, Austria, 3 LaPsyDÉ, Université Paris Cité, Paris, France, 4 Department of Neurology, Medical University of Innsbruck, Innsbruck, Austria, 5 Department of Neurology, Medical University of Vienna, Vienna, Austria, 6 Department of Psychiatry, Psychotherapy and Psychosomatics, Division of Psychiatry I, Medical University of Innsbruck, Innsbruck, Austria, 7 Department of Neurology 2, Kepler University Hospital, Med Campus III, Linz, Austria, 8 Department of Neurology, Division of Neurogeriatrics, Medical University of Graz, Graz, Austria, 9 Department of Neurology, Medical University of Vienna, Vienna, Austria, 10 Department of Psychiatry and Psychotherapy A, General Hospital, Hall, Austria

☯ These authors contributed equally to this work.
* liane.kaufmann@klinikumevb.de (LK); josef.marksteiner@tirol-kliniken.at (JM)

## Abstract

Alzheimer's disease (AD) imposes a major burden on affected individuals, their caregivers and health-care systems alike. Though quite many risk factors for disease progression have been identified, there is a lack of prospective studies investigating the interplay and predictive value of a wide variety of patient variables associated with cognitive deterioration (defined as key feature of AD progression). Study participants were patients with probable and possible AD, that were assessed at four time points over a period of two years (T1-T4). The main results were threefold: (i) over time, significant changes were observed regarding patients' cognitive functioning, activities of daily living and caregiver load (but not depression, pain, neuropsychiatric symptoms); (ii) intercorrelations between caregiver load and patients' cognitive and functional variables were high, correlation patterns remaining rather stable across time; (iii) cognitive functioning at T4 was best predicted by patients' age, sex, atrial fibrillation and activities of daily living at T1; and (iv) across all four assessment points, cognitive functioning was best predicted by time (i.e., disease duration), age, sex, activities of daily living and depression. Overall, even in early stages of AD and during a short two-year period, functional changes were significant and tightly intertwined with caregiver load, thus stressing the need to consider caregiver load when diagnosing and treating patients with AD. A novel and clinically relevant finding is that even in early stages of AD, cognitive deterioration was best predicted by a combination of patients' demographic, somatic and functional variables.

**Data Availability Statement:** The underlying data can be found at the OSF repository. The DOI is as follows: 10.17605/OSF.IO/R7HSF.

**Funding:** The work of RJ was supported by a grant from the French National Research Agency (ANR-22-CE28-0020; https://anr.fr/Project-ANR-22-CE28-0020).

**Competing interests:** The authors have declared that no competing interests exist.

## Introduction

Alzheimer´s disease (AD) is the most common form of dementia, affecting 50 to 70% of patients diagnosed with dementia [1]. Importantly, the high–and with increasing life expectancies still rising—incidence rates of AD [2,3] are associated with high public health costs as affected individuals require comprehensive interdisciplinary attention (including medical, therapeutical, palliative care).

Upon acknowledging the societal impact of AD (including the associated challenges and costs for medical diagnosis and treatment), it is not surprising that there is a vast literature on potential risk factors for AD (other than old age which previously has been identified a primary risk factor for AD) [3,4].

### Somatic and functional risk factors

Concerning *somatic risk factors* for cognitive impairment and incident dementia, findings of recent meta-analyses reveal that patients suffering from coronary heart diseases are at increased risk to develop cognitive impairment and dementia [5,6]. Moreover, there is accumulating evidence that hypertension, atrial fibrillation, diabetes mellitus and hypercholesterolemia are important risk factors for cognitive decline, eventually leading to AD [7; see also 8 and 9 for comprehensive and critical overviews of potentially modifiable risk factors for dementia]. Notably, successful treatment of vascular risk factors such as lowering high blood pressure has been reported to be significantly related to a lower risk of cognitive decline and incident dementia [10,11]. Because previous findings were targeted at examining the effects of systolic blood pressure interventions on cognitive deterioration in dementia [12], we decided to consider both systolic and diastolic blood pressure in our analyses. Importantly, as old age frequently is associated with several of the above-mentioned somatic risk factors, the negative effects of each of these disorders may overlap and synergize, thus heightening the risk of cognitive impairment and AD [7].

Beyond somatic risk factors, also *functional non-cognitive variables* have been identified as potential risk factors or factors thought to facilitate disease progression. Above all, *depression* (beyond age) has been identified as a major risk factor for the development of AD [13] and moreover, depressive symptoms and depression severity were found to facilitate and even predict the conversion from mild cognitive impairment to AD [14,15]. Recent findings of a large-scale prospective cohort-study disclosed that depression is associated with a 51% higher risk of dementia [16]. Furthermore, depression is a major burden for elderly individuals with dementia that suffer from chronic pain [17–19]. Likewise, *chronic pain* in elderly–especially in association with dementia-related cognitive changes–imposes a high burden on affected patients and their family caregivers alike [20]. Even though chronic pain is a frequent sequalae of age-associated accumulating somatic complaints (including those related to the movement apparatus) the management and treatment of pain in dementia has been hardly studied [17,18]. Although some evidence suggests that pain in the elderly is linked to *neuropsychiatric symptoms*, the evidence for this relation is controversially discussed [17,20]. However, the presence of neuropsychiatric symptoms–particularly those in the psychotic cluster–has been identified as a reliable indicator of cognitive decline (measured by the Mini-Mental-State-Examination/ MMSE) in individuals with dementia [21]. Finally, there is accumulating evidence suggesting that patients' increasing *functional dependency* (as reflected in impaired activities of daily living/ADLs) exerts a negative impact on caregiver's physical and psychological wellbeing, thus resulting in growing *caregiver load* [22,23]. Not surprisingly, accumulating evidence suggests that patients' increasing functional impairments (e.g., reduced ADLs, chronic pain and neuropsychiatric behaviors) exert direct and indirect effects on caregiver's wellbeing [23,24].

Finally, longitudinal large-scale studies found that up to half of the informal caregivers (e.g., family members, neighbors, friends) of patients diagnosed with AD report clinically significant levels of challenges related to the caregiving situation [22,25], proportions even increasing with AD progression. Interestingly, both caregiver and patient characteristics might be predictors of caregiver load. Converging evidence suggests that the most reliable indicators of patient outcome are the severity of cognitive impairment, neuropsychiatric behavioral symptoms and functional dependency [22,24,25]. Moreover, risk factors for developing caregiver load included higher age, female sex and living together with the patient [24,25]. Clearly, the presence of neuropsychiatric symptoms (that can be differentiated into behavioral, emotional, and psychotic behaviors [21]) and/or higher functional dependency (e.g., increasing need for support to manage daily affairs) is unequivocally associated with an increasing impact on caregivers' physical and mental health [22,24–26]. Hence, the examination of the interplay between caregiver load and patient-related measures—especially during early stages of AD—is of utmost interest.

### Study aims

*The focus of the present prospective study* is on demographic (age, sex, and education) and somatic potential risk factors (atrial fibrillation, coronary heart disease, hypercholesterolemia, hypertension, diabetes, diastolic and systolic blood pressure) as well as non-cognitive functional variables frequently related to AD (functional dependency as indicated by decreasing activities of daily living/ADLs, depression, pain, and neuropsychiatric symptoms). Moreover, upon acknowledging findings suggesting a correlation between caregiver load and AD patients' health status [22,24,26], also subjective caregiver load was introduced into our longitudinal analyses.

In the present study, we report data from the Prospective Registry on Dementia (PRODEM) in Austria. The PRODEM is a prospective cohort study that followed patients with AD and related dementias over a period of two years across four assessment points (i.e., baseline, six months after baseline as well as one and two years after baseline). The main aims of the present longitudinal study were as follows: **(i)** to investigate early-stage disease progression of AD (indexed by patients' functional -including cognitive- variables), **(ii)** to examine correlation patterns and correlation strengths between patients' functional -including cognitive- variables and caregiver load across time, and finally **(iii)** to identify the predictive value of patients' demographic, somatic and functional risk variables at baseline on patients' cognitive functioning at the last follow-up examination as well as across all four assessment points (see Fig 1).

## Materials and methods

### Ethics statement

Ethical approval was obtained from the ethics committees of the following participating centers: Medical University of Graz, Medical University of Innsbruck, Medical University of Vienna, Konventhospital Barmherzige Brüder Linz as well as the Provinces of Upper Austria, Lower Austria and Carinthia. Please note that in Austria, it is sufficient to obtain ethical approval from the leading study center. The leading study center in the present study was the Medical University of Graz, thus, ethical approval was obtained from the ethics committee of the Medical University of Graz (approval number 19–135 ex 07/08) and accepted by the ethics committees of the other participating study centers.

### Study participants

All participating patients and their caregivers were recruited via the multi-centric cohort study "Prospective Dementia Registry Austria (PRODEM-Austria)" of the Austrian Alzheimer

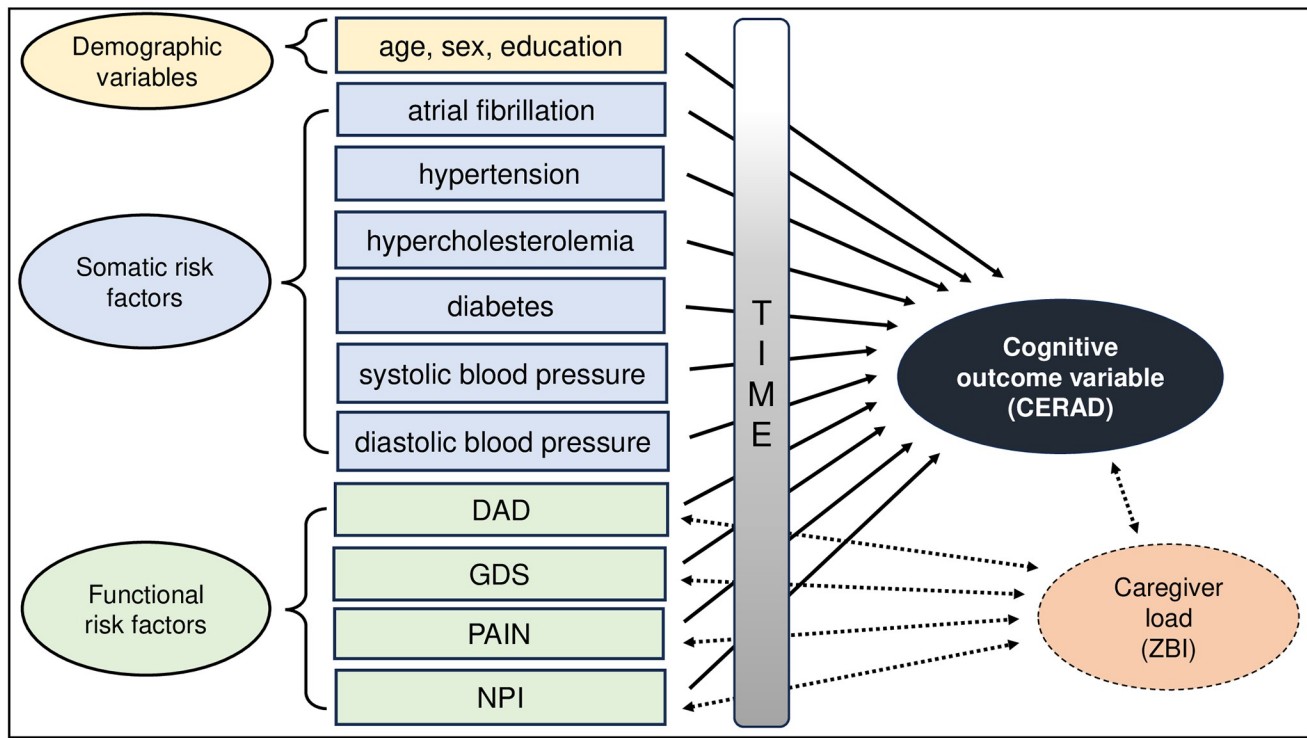

**Fig 1. Schematic representation of the study variables and study aims (one-directional solid arrows indicating potential unidirectional associations between patients' risk factors and the cognitive outcome measure as indexed by the CERAD, bidirectional dashed arrows indicating potential interrelations between patients' functional variables and caregiver load as indexed by the ZBI).** *Abbreviations:* CERAD = Consortium to Establish a Registry for Alzheimer's Disease; DAD = Disability Assessment for Dementia (indexing activities of daily living/ADLs); GDS = Geriatric Depression Scale; Pain = Pain scale focusing on pain during activity; NPI = Neuropsychiatric Inventory; ZBI = Zarit Burden Inventory. *Notes:* Somatic risk factors were recorded at baseline only (T1), while cognitive functioning (indexed by CERAD), functional non-cognitive risk factors and caregiver load were examined at all four assessment points (i.e., from T1 to the last follow-up two years after baseline/T4).

Society. Data collection for the present study started on January 23, 2009 and ended April 21, 2017. Written informed consent was obtained from all study participants. All participating patients and caregivers underwent a comprehensive clinical examination at baseline (T1) and were invited for regular clinical routine follow-up examinations six month after baseline (T2) as well as one and two years after baseline (T3 and T4, respectively).

**Inclusion criteria.** Inclusion criteria were as follows: living still at home, availability of a family caregiver who could provide information on the patient's neuropsychiatric symptoms and health condition. Excluded were patients with reduced capacity to consent and/or who were unable to sign an informed consent or if co-morbidities were likely to preclude study termination. Importantly, reduced capacity to consent was an exclusion criterion at all time points (i.e., T1 to T4). In our sample, no single patient had to be excluded at the follow-up visits because of severe cognitive deterioration yielding reduced capacity to consent and/or difficulties to understand the medical and neuropsychological procedures associated with the follow-up visits. The participating study sites were federal or provincial hospitals specialized in neurology or psychiatry and located in six of the nine provinces in Austria. Upon applying the latter criteria, the study sample consisted of 578 patients diagnosed with possible or probable AD according to NINCDS-ADRDA criteria [27,28]. Due to missing data (i.e., not all patients completed all tests at the four assessment points) we decided to include only patients into the analyses for whom we had data from the cognitive outcome measure (CERAD) and at least

**Table 1. Patient and caregiver characteristics as well as patients' MMSE scores (as a proxy for cognitive functioning/deterioration).**

| | T1 (baseline) | T2 (six-month follow-up) | T3 (one-year follow-up) | T4 (two-year follow-up) |
|---|---|---|---|---|
| **Patients** | | | | |
| Sample size: total patient group (possible/probable AD) | N = 500 (170/330) | N = 352 (115/237) | N = 289 (93/196) | N = 169 (50/119) |
| Age: years (SD) | 76.68 (7.68) | 76.89 (7.47) | 77.60 (7.56) | 78.06 (7.29) |
| Sex: female/male | 297/202 | 207/144 | 166/122 | 100/69 |
| Education: years (SD) | 11.14 (2.29) | 11.20 (2.35) | 11.27 (2.42) | 11.12 (2.40) |
| MMSE: raw score (SD), subtest CERAD/max. 30 raw points | 22.29 (3.96) | 21.59 (4.15) | 20.71 (5.05) | 19.28 (5.80) |
| **Caregivers** | | | | |
| Sample size | N = 493 | N = 348 | N = 288 | N = 168 |
| Relationship to patient: spouse/adult-child/other | 227/172/94 | 165/120/63 | 139/97/52 | 92/53/23 |
| Age: years (SD) | 60.32 (14.33) | 61.11 (14.54) | 61.89 (14.97) | 63.75 (15.01) |
| Sex: female/male | 335/159 | 240/108 | 200/87 | 119/50 |

*Abbreviations*: MMSE = Mini Mental Status Examination; SD = standard deviation.

*Notes*: Due to missing values, sample sizes of patients and caregivers do not perfectly match (maximum difference value being n = 7 at T1; n = 4 at T2; n = 1 at each T3 and T4).

one non-cognitive functional variables at each assessment point (i.e., T1 to T4; see the section"Functional non-cognitive measures" below). Hence, at baseline, the initial study sample comprised N = 500 patients.

**Data acquisition.** First, all patients underwent clinical examinations that involved the acquisition of demographic data (Table 1) and results of functional (including cognitive) tests. Moreover, patients' medical records were used to obtain information regarding somatic risk factors (Table 2). As outlined in the introduction, the variable selection was based on the literature and thus, is selective rather than comprehensive. In addition, family caregivers were asked to assess patients' functional dependency (i.e., activities of daily living/ADLs) and potential behavioral changes (i.e., neuropsychiatric symptoms, known to be frequently associated with dementia) and finally, were asked to report on their caregiver load (see Table 1 for caregivers' demographic data).

**Table 2. Somatic risk variables for the development of dementia.** Reported are the number of cases with a specific somatic diagnosis (n reported/n not reported).

| Somatic variables | T1 N = 500 (n reported/n not reported) |
|---|---|
| Atrial fibrillation | 47/412 |
| Coronary heart diseases | 74/383 |
| Hypercholesterolemia | 206/248 |
| Hypertension | 289/176 |
| Diabetes | 65/399 |
| Systolic blood pressure in mmHg: M (SD) | 140.18 (25.84) |
| Diastolic blood pressure in mmHg: M (SD) | 82.01 (12.41) |

*Abbreviations*: mmHg = millimeters of mercury; SD = standard deviation.

*Notes*: Due to missing values, some of the sample sizes reported at specific somatic variables might be somewhat lower than the reported total sample size (N = 500).

**Table 3. Comparison of patients who completed the study until the two-year follow-up visit (T1 to T4) and patients who dropped out after baseline (no follow-up visit).**

| Demographic data and somatic variables at baseline | Mann-Whitney Tests[1] / Chi-Squared Tests[2] | | |
|---|---|---|---|
| | Patients who remained in study (T1 to T4) N = 169[§] | Drop-out N = 331[§] | *p* two-tailed |
| Age: years (SD)[1] | 76.06 (7.29) | 76.99 (7.82) | 0.214 |
| Sex: female/male[2] | 100/69 | 199/134 | 0.976 |
| Education: years (SD)[1] | 11.12 (2.40) | 11.12 (2.23) | 0.558 |
| MMSE: raw score (SD)[1] | 22.65 (3.65) | 22.10 (4.14) | 0.399 |
| ZBI (indexing caregiver load): raw score (SD)[1] | 19.39 (14.07) | 16.28 (11.78) | 0.031* |
| Atrial fibrillation[2] | 15/149 | 33/265 | 0.624 |
| Coronary heart diseases[2] | 31/132 | 44/253 | 0.301 |
| Hypercholesterolemia[2] | 88/79 | 120/170 | 0.025* |
| Hypertension[2] | 112/56 | 179/121 | 0.162 |
| Diabetes[2] | 27/139 | 38/262 | 0.350 |
| Systolic blood pressure: mmHg (SD)[2] | 142.67 (22.38) | 138.83 (27.34) | 0.311 |
| Diastolic blood pressure: mmHg (SD)[2] | 84.41 (12.54) | 80.80 (12.22) | 0.011* |

*Notes*:

[1]*p*-values were determined by Mann-Whitney Tests when analyzing interval scaled data);

[2]*p*-values were determined by Chi-Squared Tests when analyzing dichotomous variables (e.g., sex and some somatic variables);

[§]provided are the maximal sample sizes (however, sample sizes might be somewhat lower for specific somatic variables due to missing data);

*significant results at $p < 0.05$ (please note that group differences were no longer statistically significant after applying Bonferroni alpha-corrections, $p_{min} = 0.121$).

*Abbreviations*: MMSE = Mini Mental Status Examination; ZBI = Zarit Burden Inventory; mmHg = millimeters of mercury; SD = standard deviation.

**Drop-out analysis.** Across the four study time points, the number of participating patients decreased from n = 500 at baseline (T1) to n = 169 at T4 (i.e., two-year follow-up; Table 1). Though this drop-out rate is rather high, it is according to expectation as in clinical longitudinal studies, patients are frequently lost due to translocation, change of medical treatment/hospital, disease progression hampering study compliance, etc. [29–31].

Importantly, the *drop-out analysis* disclosed no significant differences between patients who were lost to follow-up and those who remained in the longitudinal study regarding demographic variables (i.e., age, sex, and education), a cognitive screening test (i.e., MMSE) and all but two somatic variables (i.e., atrial fibrillation, coronary heart disease, hypertension, diabetes, and systolic blood pressure). The only significant group differences emerged as regards hypercholesterolemia and diastolic blood pressure (Table 3). Moreover, significant group differences between patients who dropped out and those who remained in the study were observed regarding caregiver load (indexed by the Zarit Burden Inventory/ZBI).

## Cognitive and non-cognitive functional measures (tests and questionnaires)

The present longitudinal study comprised four assessment times (i.e., baseline followed by three follow-up examinations). Please note that for all measures (i.e., the cognitive outcome measure CERAD and the non-cognitive functional variables) solely the total scores were entered into the data analyses.

**Overall cognitive functioning.** Overall, cognitive functioning was assessed by asking all patients to complete the CERAD neuropsychological test battery (Consortium to Establish a

Registry for Alzheimer's Disease [32]) which includes subtests measuring semantic verbal fluency, a confrontation naming test (i.e., modified Boston Naming Test), a verbal memory test (tapping word list learning, word list immediate and delayed recall, word list recognition), a constructional praxis test (copying figures), a figural memory test (delayed recall) as well as the MMSE [33]. Out of these measures, we calculated a CERAD total score (age and education corrected z-scores [34]). The CERAD total score was used as cognitive outcome measure and served as a proxy for disease progression. Overall, higher z-scores indicate better cognitive performance (and vice versa).

**Activities of daily living.** Activities of daily living (ADLs) were assessed by using the scale Disability Assessment for Dementia (DAD) [35,36]. The DAD requires caregivers to report on patients' basic and instrumental activities of daily living as well as leisure activities over the last two weeks. Overall, the DAD consists of 40 items that assess whether a specific ADL is still preserved or not (YES vs NO answer, respectively). A third answer category is reserved for items that are not applicable (NA). Please note that NA answers are not considered and thus, reduce the maximum number of items. The DAD total score is provided as a percentage (number of YES items x 100 divided by the maximum number of items). In the present study, the total DAD score was used to measure the patients' ADLs, with lower scores indexing higher levels of functional dependency (and higher scores indicating higher levels of functional independency).

**Depressive symptoms.** Depressive symptoms were examined by using the short version of the Geriatric Depression Scale (GDS-15) [37,38]. The GDS short version consists of 15 items that require a YES or NO answer (max. score 15). Higher scores indicate a more pronounced depressive symptomatology.

**Pain.** Pain was assessed using a structured pain questionnaire specifically developed for geriatric patients (developed and registered by the German Pain Society) [39]. The original pain interview consists of questions requiring the patient to provide information regarding pain localization (including number of body parts affected by pain), pain intensity and functional impairments caused by pain (i.e., hampering patients' ADLs). Based on the literature suggesting that pain during activity is a frequent complaint among the elderly, both with and without dementia [40,41], we chose to focus our analyses on questions tapping pain during activity. Overall, pain during activity was assessed by four items (taken from Basler [39]). Participants were required to answer YES if they were still able to perform the specific activity, or alternatively, to answer NO if they were unable to perform the activity due to pain or due to other reasons. The total pain score was derived by adding the NO answers due to pain only, thus yielding a total score of 4, with higher scores indicating more pain.

**Neuropsychiatric symptoms.** Neuropsychiatric symptoms were evaluated using the Neuropsychiatric Inventory (NPI) [42]. The NPI assesses 12 subdomains of behavioral functioning (i.e., delusions, hallucinations, agitation/aggression, depression/dysphoria, anxiety, irritability/lability, euphoria, apathy, disinhibition, aberrant motor behavior, night-time behavioral disturbances, appetite and eating abnormalities). In the present study, we used the total NPI score, that is derived by multiplying symptom frequency (scored on a 4-point scale ranging from 0 (never) to 4 (at least once per day)) with symptom severity (scored on the 3-point scale mild, moderate and severe) for each of the 12 subdomains separately. Hence, the maximum raw score is 144 (12 symptoms x 4 x 3), with higher scores indicating more neuropsychiatric symptoms.

**Caregiver load.** Caregiver's subjective feelings of challenges related with the caregiving situation were measured by using the German-language version of the Zarit Burden Inventory (ZBI) [43,44]. The ZBI consists of 22 items related to patients' behavioral or functional impairments and/or the care situation. Items were answered on a 5-point Likert scale (ranging from

0 to 4). Thus, the total score ranged from 0 to 88, with higher scores indicating stronger feelings of burden.

## Statistical methods

All analyses were conducted by using the statistic software R (version 4.3.0) [45]. For all analyses, results were considered significant if $p < 0.05$. At each timepoint, we included all patients who had a CERAD score and, moreover, a valid measurement of at least one additional functional test (i.e., DAD, GDS, PAIN, NPI).

**Measures.** For all functional tests (i.e., cognitive outcome measure CERAD as well as all non-cognitive functional tests), we used the total raw scores as dependent variables. The dependent variables of the somatic patient variables were–alike the functional patient variables—either interval scaled (i.e., diastolic and systolic blood pressure) or dichotomous (i.e., yes/no: atrial fibrillation, coronary heart disease, hypercholesterolemia, hypertension, diabetes). Likewise, demographic variables were interval scaled (i.e., age, education in years) or dichotomous (sex).

**Statistical analyses.** Beyond reporting on the descriptive results of the functional variables (including the CERAD total score used as a cognitive outcome measure, see data reported in S1 Fig in S1 File), we calculated linear mixed models—using the *lme4* package (version 1.1.34 [46]; time being included as a fixed effect, while patient-ID was entered as a random effect) aiming at investigating whether functional changes over time were significant (Table 4).

In a next step, Pearson correlation analyses were conducted (by using the *Hmisc* package, version 5.0.1 [47]) to examine whether correlation strengths and patterns between the functional (cognitive and non-cognitive) variables changed across the four assessment points (Table 5).

Finally, linear regression analyses were performed to identify potential causal relationships between the cognitive outcome measure and demographic, somatic and functional patient variables (see Tables 6 and 7). As depicted in Table 6, a bidirectional stepwise linear regression analysis was conducted (using the *caret* package, version 6.0.94 [48]) to examine whether the dependent variable (i.e., cognitive outcome measure as indexed by the CERAD total score at T4) can be predicted by any of the predefined independent variables at T1 (i.e., patients' demographic, somatic and functional risk variables). Predictors were sequentially added and removed to find the set of predictors best suited for variance explanation. Moreover, 5-fold

**Table 4. Results from linear mixed models reflecting linear development across time of the cognitive outcome measure (CERAD), patients' functional variables (DAD, GDS, PAIN, NPI) and caregiver load (ZBI).** Values in squared brackets indicating 95% confidence intervals.

| Name of scale | *SDindividual* | *SDresidual* | Intercept[a] | Time [T1, T2, T3, T4][b] |
|---|---|---|---|---|
| CERAD | 5.24 [4.88, 5.59] | 2.56 [2.43, 2.68] | **-10 [-10.49, -9.51]** | **-1.27 [-1.51, -1.03]** |
| DAD | 21.31 [19.76, 22.87] | 13 [12.35, 13.64] | **74.46 [72.37, 76.56]** | **-10.68 [-11.89, -9.46]** |
| GDS | 1.74 [1.59, 1.90] | 1.61 [1.52, 1.69] | **2.58 [2.39, 2.78]** | >-.01 [-0.15, 0.14] |
| PAIN | .91 [0.80, 1.02] | 1.25 [1.19, 1.31] | **0.92 [0.79, 1.04]** | -0.1 [-0.21, 0.01] |
| NPI | 13.52 [12.48, 14.56] | 9.91 [9.42, 10.39] | **13.08 [11.68, 14.48]** | 0.56 [-0.36, 1.48] |
| ZBI | 11.92 [11.00, 12.85] | 8.39 [7.97, 8.82] | **18.32 [17.10, 19.54]** | **3.8 [3.01, 4.58]** |

*Notes*: For all variables, total scores are provided (as only total scores were entered into data analyses); significant results ($p < .05$) are printed in bold letters.

*Abbreviations*: CERAD = Consortium to Establish a Registry for Alzheimer's Disease; DAD = Disability Assessment for Dementia; GDS = Geriatric Depression Scale;

Pain = Pain scale focusing on pain during activity; NPI = Neuropsychiatric Inventory; ZBI = Zarit Burden Inventory.

[a]The intercept indicates the baseline value and $SD_{individual}$ indicates the variation of that estimated baseline value.

[b]Time is indicated as follows: T1 = baseline assessment, T2 = follow-up six months after baseline, T3 = one-year follow-up, T4 = two-year follow-up.

**Table 5. Pattern of correlation strengths between functional variables, separately reported for the four time points.**

| | | CERAD | DAD | GDS | PAIN | NPI | ZBI |
|---|---|---|---|---|---|---|---|
| **DAD** | T1 | 0.23**$ | 1 | | | | |
| | T2 | 0.30**$ | 1 | | | | |
| | T3 | 0.36**$ | 1 | | | | |
| | T4 | 0.46**$ | 1 | | | | |
| **GDS** | T1 | -0.08 | -0.12** | 1 | | | |
| | T2 | -0.13* | -0.14* | 1 | | | |
| | T3 | -0.13* | -0.01 | 1 | | | |
| | T4 | -0.12 | -0.11 | 1 | | | |
| **PAIN** | T1 | 0.02 | 0.06 | 0.00 | 1 | | |
| | T2 | 0.02 | 0.17** | -0.02 | 1 | | |
| | T3 | 0.05 | 0.10 | -0.01 | 1 | | |
| | T4 | 0.05 | 0.13 | 0.01 | 1 | | |
| **NPI** | T1 | -0.14** | -0.41**$ | 0.24**$ | -0.06 | 1 | |
| | T2 | -0.08 | -0.39**$ | 0.19**$ | -0.06 | 1 | |
| | T3 | -0.08 | -0.44**$ | 0.20** | -0.04 | 1 | |
| | T4 | -0.16* | -0.43**$ | 0.07 | -0.02 | 1 | |
| **ZBI** | T1 | -0.14** | -0.50**$ | 0.11* | 0.15** | 0.45**$ | 1 |
| | T2 | -0.14* | -0.54**$ | 0.09 | -0.03 | 0.44**$ | 1 |
| | T3 | -0.30**$ | -0.57**$ | 0.12* | 0.03 | 0.42**$ | 1 |
| | T4 | -0.35**$ | -0.54**$ | 0.21** | -0.09 | 0.44**$ | 1 |

*Notes*:

*$p < 0.05$;

**$p < 0.01$; significant results are printed in bold letters;

$significant correlations after Bonferroni alpha correction.

*Abbreviations*: T1 = baseline; T2 = follow-up six month after baseline; T3 = follow-up one year after baseline; T4 = follow-up two years after baseline;

CERAD = Consortium to Establish a Registry for Alzheimer's Disease; DAD = Disability Assessment for Dementia; GDS = Geriatric Depression Scale; Pain = Pain scale focusing on pain during activity; NPI = Neuropsychiatric Inventory; ZBI = Zarit Burden Inventory.

**Table 6. Optimal model (as disclosed by stepwise regression analyses) to predict cognitive functioning at the last follow-up (T4) upon using patients' variables at baseline (T1).**

| Predictors (T1) | CERAD (T4) | | |
|---|---|---|---|
| | *Estimates* | *CI$_{95\%}$* | *p* |
| (Intercept) | -32.64 | -44.90, -20.37 | <0.001 |
| Age | 0.26 | 0.11, 0.40 | <0.001 |
| Sex | -2.16 | -4.21, -0.11 | 0.039 |
| Atrial fibrillation | 4.31 | 0.64, 7.98 | 0.022 |
| DAD | 0.05 | 0.01, 0.10 | 0.020 |
| $R^2$ / Adjusted $R^2$ | 0.160 / 0.137 | | |

*Notes*: $CI_{95\%}$ = 95% confidence interval; significant results are printed in bold letters.

*Abbreviations*: CERAD = Consortium to Establish a Registry for Alzheimer's Disease; DAD = Disability Assessment for Dementia; GDS = Geriatric Depression Scale; Pain = Pain scale focusing on pain during activity;

NPI = Neuropsychiatric Inventory; ZBI = Zarit Burden Inventory.

**Table 7. Linear mixed model depicting the relationships between patient's functional variables and cognitive functioning across time.**

| Predictors (T1 –T4) | CERAD (T1 –T4) | | |
|---|---|---|---|
| | Estimates | CI | p |
| (Intercept) | -24.04 | -29.44, -18.64 | **<0.001** |
| time | -0.80 | -1.07, -0.54 | **<0.001** |
| Age | 0.19 | 0.13, 0.25 | **<0.001** |
| Sex | -2.29 | -3.20, -1.37 | **<0.001** |
| Education | -0.09 | -0.28, 0.11 | 0.390 |
| DAD | 0.06 | 0.05, 0.07 | **< .0001** |
| GDS | -0.11 | -0.21, -0.01 | **0.028** |
| PAIN | 0.00 | -0.13, 0.13 | 0.961 |
| NPI | 0.00 | -0.02, 0.02 | 0.972 |
| **Random Effects** | | | |
| $\sigma^2$ | 5.80 | | |
| $\tau_{00 \ ident}$ | 20.53 | | |
| ICC | 0.78 | | |
| Conditional $R^2$ / Marginal $R^2$ | 0.814 / 0.157 | | |

*Notes*: CI = 95% confidence interval, significant results are printed in bold letters.

*Abbreviations*: CERAD = Consortium to Establish a Registry for Alzheimer's Disease; DAD = Disability Assessment for Dementia; GDS = Geriatric Depression Scale; Pain = Pain scale focusing on pain during activity; NPI = Neuropsychiatric Inventory; ZBI = Zarit Burden Inventory.

cross-validation was used to ensure that the result from the stepwise regression would be unbiased and reliable. Finally, the regression model with the best fit to our data (i.e., the lowest 'root mean square error of approximation'/RMSEA) was then used for the following analyses. The final model was checked for collinearity and potential interactions between these predictors. Importantly, our findings from stepwise regression analysis seem to be robust and not biased by any potential interaction effects between the predictor variables.

In addition (see Table 7), by using linear mixed models (treating time as fixed effect and patient-ID as random effect) we aimed at investigating whether, across the four assessment points, functional risk factors were predictive of cognitive functioning (entered as dependent variable and indexed by the CERAD total score at each time point).

To counteract the problem of multiple comparisons (i.e., type 1 error), we applied Bonferroni corrections to the drop-out and the correlation analyses.

Upon interpreting whether statistically significant results were also clinically meaningful, we used the benchmarks proposed by Cohen [49], suggesting that Pearson's correlation coefficients *r* between 0.1 and 0.3 indicate small effect sizes, those between 0.3 and 0.5 medium and those $\geq$ 0.5 large effect sizes (which correspond to Cohen's *d* of 0.2, 0.5 and 0.8, respectively). Likewise, upon interpreting the regression results (i.e., the coefficients of determination $R^2$ indicating the percentage of variance of the dependent variable explained by the independent variables), we follow Cohen's [49] suggestion proposing that $R^2$ of 0.02 reflect small, 0.13 medium and 0.26 strong effect sizes.

## Results

In the following, we will elaborate our findings along the three main research questions as outlined in the section "Study aims".

### Disease progression in the early stages of AD

To examine whether cognitive and non-cognitive functional changes across time (i.e., from the baseline assessment at T1 to the last follow-up two years later at T4) were significant and clinically meaningful, we estimated linear mixed models for each functional variable (Table 4). Interestingly, significant changes over time were observed regarding CERAD (overall cognitive functioning), thus reflecting poorer cognitive abilities with increasing disease duration (for the descriptive results, see data in S1 Fig in S1 File of the Supporting Information). Moreover, DAD (tapping ADLs) and ZBI (caregiver load) changed significantly across time. As depicted in the data in S1 Fig in S1 File, DAD scores decreased significantly over time, indicating higher functional dependency, and higher ZBI scores reflected higher caregiver load. In contrast, ratings of GDS (depressive symptoms), PAIN (tapping pain during activity), and NPI (neuropsychiatric symptoms) did not reach significance.

### Correlation between patient-related variables and caregiver load over time

The strongest and highly significant correlations were observed between caregiver load (ZBI) and patients' ADLs (DAD), neuropsychiatric symptoms (NPI) and cognitive functioning (CERAD). Although correlation patterns remained rather stable over time, correlation strengths changed somewhat across the four assessment points (Table 5, also see data in S2A-S2E Fig in S1 File). Higher caregiver load was associated with poorer ADLs (correlation coefficients > -0.50 at all assessment points), more neuropsychiatric symptoms (correlation coefficients ranging from 0.42 to 0.45), and lower cognitive functioning (correlation coefficients ranging from -0.14 to -0.35). Furthermore, as depicted in Table 5, caregiver load was significantly and positively correlated with depressive symptoms (GDS) at three out of the four time points (i.e., T1, T3, and T4). Another highly significant correlation emerged between cognitive functioning (CERAD) and ADLs. Most notably, across time the latter correlation became increasingly stronger (correlation coefficients ranging from 0.23 to 0.46; positive correlation coefficients indicating that lower CERAD scores being associated with poorer ADLs as indexed by lower DAD scores). In addition, CERAD scores were significantly correlated with patients' depressive symptoms (GDS) at T2 and T3 (correlations being non-significant at T1 and T4), reflecting that lower CERAD scores went along with higher GDS scores (i.e., more depressive symptoms). Furthermore, CERAD scores were significantly correlated with neuropsychiatric symptoms (NPI) at T1 and T4 (correlations remaining non-significant at T2 and T3), indicating that lower CERAD scores were associated with higher NPI scores (i.e., more neuropsychiatric symptoms). Furthermore, at three out of four time points (i.e., from T1 to T3), neuropsychiatric symptoms (NPI) were positively and significantly correlated with depressive symptoms (GDS). Finally, there was a correlation between pain and caregiver load at T1 ($r = 0.15$) but the correlation did not reach significance at the follow-up visits (see Table 5 for a full overview of all correlations).

### Predictive variables for cognitive deterioration

*First*, upon employing stepwise regression analysis, we sought to examine whether and which patient variables at T1 (i.e., demographic, somatic and functional non-cognitive variables) might be predictive of cognitive deterioration (i.e., cognitive functioning as indexed by CERAD score at T4). *Second*, by employing a linear mixed model, we sought to identify patient variables (i.e., demographic and non-cognitive functional risk factors) that might be significant predictors of cognitive functioning across the four follow-up visits (i.e., from T1 to T4). Please note that somatic patient variables were not entered into the latter analyses, as somatic variables were collected at T1 only.

Results of the stepwise regression analysis disclosed that patients' cognitive functioning at T4 was significantly predicted by patients' age and sex as well as atrial fibrillation and ADLs (i.e., functional dependency as indexed by DAD) at baseline (Table 6, see also data in S3 and S4A-S4D Figs in S1 File). The overall model fit disclosed an *adjusted $R^2$* = 0.137 (corresponding to a medium effect size according to Cohen [49]).

As depicted in Table 7, results of the linear mixed model disclosed a good overall fit (*marginal $R^2$* = 0.157, corresponding to a medium effect size according to Cohen [47]) and disclosed that cognitive functioning is best predicted by time (i.e., disease duration), age, sex, ADLs (i.e., functional dependency as indexed by DAD) and depressive symptomatology (GDS). More specifically, increasing time (being a proxy for disease duration/progression), high age, female sex, higher functional dependency and more severe depressive symptomatology were found to be associated with lower cognitive functioning.

## Discussion

In this study, we examined the impact of various somatic and non-cognitive functional patient-related variables on the progression of early Alzheimer's disease (AD) as measured by a cognitive outcome measure (i.e., CERAD total score). Furthermore, we investigated the relationship between these patient-related variables and the level of caregiver load. Our primary findings were as follows: *First*, significant changes in functional capacity were observed over time in relation to patients' cognitive functioning (i.e., CERAD total score [34]), patients' ADLs and caregiver load. However, no significant changes were observed in the remaining functional variables (such as depressive symptoms, pain, and neuropsychiatric symptoms) between the baseline (T1) and the final follow-up visit two years after baseline (T4). *Second*, the correlation analyses revealed a strong and consistent relationship between caregiver load and patients' cognitive and non-cognitive functional variables over the two-year study period. Notably, caregiver load was found to be significantly and positively correlated with patients' ADLs, neuropsychiatric symptoms, cognitive functioning (CERAD), and depressive symptoms) across all four assessment points. *Third*, regression analyses showed that among all patient variables collected at T1, only age, sex, atrial fibrillation, and ADLs were significant predictors of cognitive functioning at T4 (measured by the CERAD total score). Additionally, across all four timepoints, patients' cognitive functioning was best predicted by time, patients' age, sex, ADLs, and depressive symptoms. In the following, we will discuss these findings in greater detail and consider their implications for our understanding of AD and its progression.

### Disease progression during early stages of AD

Regarding cognitive decline (as reflected in changes of CERAD scores from T1 to T4), our findings contribute to the existing body of literature [1,3,4] by revealing the significant deterioration of cognitive abilities even in the early stages of AD. This decline occurs over a relatively short period of two years. Additionally, our findings disclosed notable changes in patients' ADLs, which reflect their increasing functional dependency as measured by the DAD. Furthermore, significant changes were observed in subjective feelings of caregiver load, as indicated by the ZBI, throughout the two-year follow-up. While not entirely unexpected, our findings disclose that even in the early stages of the disease, significant decreases in ADLs are associated with significant increases in caregiver load. Therefore, our findings, in conjunction with the correlation analyses discussed in the subsequent section, further validate the close relationship between patients' non-cognitive functional dependency and caregiver load [22,24–26,50]. Furthermore, the seemingly insubstantial, yet quantifiable augmentation of neuropsychiatric

symptoms (indexed by the NPI) indicates that increasing disease duration is correlated with an amplified manifestation of neuropsychiatric symptoms [21,51]. Our findings partially resemble prior research indicating significant changes over time in NPI ratings related to one of the three NPI subsyndromes (specifically, the psychotic subsyndrome), while no significant alterations were observed over time in the behavioral and emotional NPI subsyndromes [21]. However, the total NPI score was not provided in the aforementioned study, thereby rendering a direct comparison unfeasible. Conversely, and somewhat unexpectedly, patients' self-assessments of pain levels remained relatively low and displayed negligible fluctuations over time. The reasons for this phenomenon remain ambiguous and may range from diminished pain perception to improved pain management techniques [18–20].

## Correlation between patient-related variables and caregiver load

As depicted in Table 5, we observed the strongest and highly significant correlations between caregiver load and patients' ADLs, neuropsychiatric symptoms and cognitive functioning. Higher caregiver load was associated with poorer ADLs, more neuropsychiatric symptoms, and lower cognitive functioning. Over a two-year period (i.e., at T1, T3 and T4), caregiver load was consistently correlated with depressive symptoms. Strong interrelations were also found among patients' cognitive and non-cognitive functional variables. Poorer ADLs were consistently correlated with lower cognitive functioning and more neuropsychiatric symptoms. Thus, our findings highlight the significant and stable interplay between caregiver load and patients' functional risk factors during early stages of AD. This emphasizes the importance of systematic evaluations of cognitive and non-cognitive functional impairments in clinical practice. In addition, our results align with previous studies indicating a strong link between caregiver load and patients' behavioral symptoms, especially disruptive behaviors as measured by the NPI [26], as well as non-cognitive functional decline, including impaired ADLs [22,25]. However, a unique feature of our study is its prospective design, covering a two-year follow-up period, and examining somatic and functional non-cognitive risk factors as potential predictor variables for cognitive decline. We specifically focus on the early stages of AD, conducting a combined and longitudinal investigation of caregiver load and fundamental non-cognitive patient functions that affect AD progression, such as ADLs, depressive symptoms, neuropsychiatric symptoms, and pain related to activity.

Moreover, our results support the idea that a systematic informant-based assessment of functional deficits serves as a valuable clinical indicator of early cognitive deterioration [52–55]. The recognition of impaired ADLs as a significant phenotypic marker for distinguishing minor from major neurocognitive disorders (DSM-V [56]) underscores the importance of conducting regular ADL assessments in clinical settings.

## Predictive variables for cognitive deterioration

Our primary objectives were to identify predictors of cognitive deterioration (CERAD at T4) by using stepwise regression at T1. Additionally, a linear mixed model was employed to discern predictors of cognitive functioning across all four assessment points over time. The stepwise regression analysis yielded a well-fitted model (*adjusted $R^2$* = 0.137, indicating a medium effect size per Cohen's benchmarks [49]; see Table 6). Approximately 14% of the variance in patients' cognitive functioning (at T4) could be explained by four predictors identified two years earlier (at T1): higher age, female sex, the presence of atrial fibrillation, and poorer ADLs were all associated with poorer cognitive functioning. Furthermore, the linear mixed model results indicated that patients' cognitive functioning over time (T1 to T4) is best predicted by time, age, sex, ADLs, and GDS (Table 7). The model showed a good fit (*marginal $R^2$* = 0.157, a

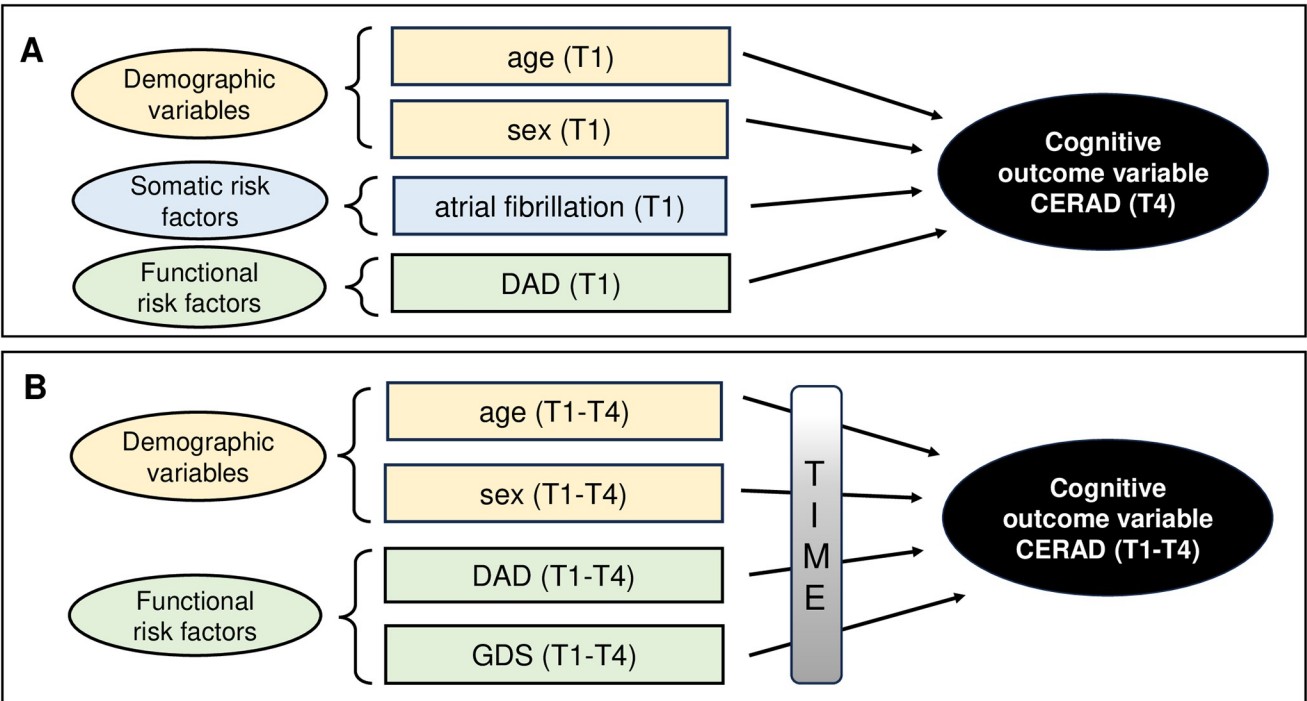

**Fig 2. Schematic representation of the results of the regression analyses.** (A) Upper panel showing the findings of the stepwise regression investigating the predictive value of all patient variables at T1 (demographic, somatic and functional) on cognitive functioning at T4. (B) Lower panel showing the findings of the linear mixed model investigating the relationship between demographic and functional patient variables on cognitive functioning across all four time points. *Abbreviations*: CERAD = Consortium to Establish a Registry for Alzheimer's Disease; DAD = Disability Assessment for Dementia (indexing activities of daily living/ADLs); GDS = Geriatric Depression Scale.

medium effect size per Cohen [49]), suggesting that approximately 16% of the variance in patients' cognitive functioning can be explained by increasing time (a proxy for disease progression), higher age, female sex, poorer ADLs (DAD), and more severe depressive symptoms (GDS). Notably, this analysis did not include somatic risk factors that were collected at T1 only (see Fig 2 for a schematic representation of the results of the two regression analyses). Overall, our model fits (suggesting medium effect sizes according to Cohen [49]) are good, especially upon acknowledging that our findings are complex and influenced by many factors, some of which we were able to assess, while we did not examine other factors that previously have been reported to influence cognitive deterioration such as lifestyle factors (physical activity, nicotine and alcohol consumption, cognitive (in)activity etc. [8,9]. Hence, although we entered quite many potential predictor variables to our stepwise regression model initially, our model might not be complete as we were not able to include all potentially relevant predictor factors for cognitive deterioration. Further potential reasons for our–at first glance—rather modest model fit are the following: (i) our prospective study aimed at investigating disease progression in early-stage dementia over a relatively short period of two years (which is very brief upon considering that dementia per definition is a slowly progressive neurodegenerative disease [1–4,57]); (ii) the high drop-out rate clearly reducing our model fit; and (iii) some of the measures and tests used were observational and/or based on subjective patient/caregiver reports and thus, possibly less sensitive than objective assessment tools. However, variance explanations of 14% and 16% (see Tables 6 and 7, respectively) are quite remarkable upon acknowledging that the functional changes were observed in early stages of a neurodegenerative disease and within a short time span of two years.

**What is the added value of the latter findings and how do they contribute to a better understanding of AD?.** Upon acknowledging the complex nature of our research questions, it does not come as a surprise that also our findings are complex and far from being comprehensive. However, our prospective multi-center study is among the first to simultaneously examine patients' demographic, somatic and functional variables as potential predictor variables for disease deterioration (indexed by the CERAD) in patients with early-stage AD. Furthermore, we aimed at elucidating the developmental trajectories of cognitive and non-cognitive functional variables across time (i.e., four assessment points spanning two years). Our findings are significant as they clearly contribute towards a deeper understanding of AD. Above all, our findings might inform future studies targeted at elucidating the complex interplay of patient and caregiver-related predictor variables. For instance, our findings provide a starting point for generating more specific hypotheses. Future studies might utilize objective rather than subjective test measures that -among others- may facilitate cross-national comparisons and, thus, enhance the reliability and generalizability of respective research findings.

Regarding predictive demographic factors, our results identified age and sex (but not education) as significant predictors of cognitive functioning or deterioration. While age is widely recognized as the primary risk factor for developing AD [1,3,4], few studies have reported findings regarding sex differences (see [58] for a comprehensive review). Most interestingly, the findings of our regression analysis disclose that, beyond age, female sex appears to be a risk factor for cognitive deterioration (being considered as a proxy for AD progression). This contribution adds to the current literature by emphasizing the importance of considering sex as a factor in understanding AD risk. On the other hand, education failed to emerge as a predictor of cognitive functioning or deterioration in our prospective study. Though converging evidence suggests that low education increases the risk of dementia (for the findings of a respective meta-analysis, see [59], our results propose that education seems less important for disease progression if other relevant risk factors are considered simultaneously (for similar findings see [57]). These results align with the idea that, although educational attainment contributes to individual differences across the lifespan, it might not necessarily serve as a protective factor against cognitive decline [59,60].

Regarding the somatic risk factors assessed at T1, only atrial fibrillation remained a significant predictor of cognitive functioning at T4 (according to the regression analysis), while none of the other predefined factors (including coronary heart disease, hypercholesterolemia, hypertension, diabetes, systolic and diastolic blood pressure) were found to be significant predictors. Consistent with our findings, there is accumulating evidence supporting the association between atrial fibrillation and cognitive impairment [61,62]. This link persists even in early-onset dementia [63]. Furthermore, several studies have shown that atrial fibrillation not only increases the risk of vascular dementia but also the risk of AD. Such as, the findings of a meta-analysis published in 2021 disclose that patients with atrial fibrillation had a 60% higher risk of incident all-cause dementia, a 40% higher risk of incident AD, and a 70% higher risk of incident vascular dementia [64]. Most likely, the relationship is multifactorial in nature, involving inflammatory, hemodynamic and/or genetic components. As noted by Rivard et al. [62], prospective large-scale studies are required to establish the causal effects of this relationship. Filling this gap, our prospective study found that even in the early stages of the disease, and beyond other somatic risk factors, atrial fibrillation should be considered a significant and clinically relevant risk factor for cognitive deterioration and, consequently, AD progression.

## Limitations

Beyond its strengths (i.e., large prospective cohort study comprising a well-defined sample of patients diagnosed with probable and possible AD, targeted at investigating potential

predictors of cognitive decline by including somatic and functional patient variables as well as caregiver variables), the present study also has potential limitations. Importantly, our findings might not be readily generalizable to a broader population as our study participants comprised solely Austrian patients with probable and possible AD. Indeed, it has been claimed previously that studies aiming at identifying potential risk factors for dementia incidence and decline are more prevalent in low- and middle-income countries compared to high-income countries (because the former bear a higher risk of potentially modifiable risk factors for dementia [8, also see 3, 60]).

A further potential limitation of the present study are the rather high drop-out rates (i.e., sample size decreasing from initially n = 500 participating patients to n = 169 at the last follow-up visit). Nonetheless, drop-out rates in prospective studies are frequently reported to be considerable [29–31]. A closer look at the reasons for drop-out disclosed that the missing data are not missing at random (MNAR) but are likely correlated with multiple patient and caregiver parameters. Upon acknowledging previous findings revealing that test results can be easily biased by imputing large amounts of MNAR data (i.e., data that are not missing at random [65]), we decided against the multiple imputation of our drop-out data to avoid biasing our results.

Moreover, some of the functional variables rest on patient and caregiver reports (e.g., pain and neuropsychiatric symptoms, respectively) which are subjective in nature and are characterized by low inter-rater reliability. To obtain objective and precise assessment results, technology-assisted assessment tools seem to be a promising avenue for future research endeavors [66,67].

Finally, though our study included quite many potential predictor variables (i.e., somatic and functional patient variables) that were regressed against the cognitive outcome variable (indexed by the CERAD), these variables were not comprehensive. In other words, we were not able to include lifestyle factors such as physical activity, nicotine and alcohol use as potential predictors to our study, although it has been argued that these factors should be considered as modifiable risk factors for dementia [8,9]. Though we did collect patients' nicotine and alcohol use (but not physical activity) in our prospective multi-center study, these data were very patchy and thus, could not be included in our analyses.

## Conclusion

To summarize, our novel findings, derived from a two-year study in early-stage AD, can be categorized into three main points: (i) significant changes in both cognitive and non-cognitive patient variables over this short time period; (ii) highly significant and stable interrelations between caregiver load and functional patient variables (cognitive functioning, ADLs, neuropsychiatric symptoms, and depressive symptoms); and (iii) the best predictors of cognitive functioning at T4 being high age, female sex, atrial fibrillation diagnosis, and poor ADLs at T1. (iv) Additionally, across all assessment points, and beyond age, sex, and ADLs, also time (indicating disease duration/progression) and depressive symptoms significantly contributed to variance explanation of patients' cognitive functioning (CERAD total score). These findings underscore the importance of a comprehensive treatment approach, considering both patient and caregiver variables, in the diagnosis and treatment of early-stage AD. Such as, our findings disclosing strong correlations between caregiver load and various patient-related measures strongly suggest that the clinical management of early-stage AD should be targeted at the patient-caregiver dyad (instead of solely focusing on the patient). Moreover, our regression findings disclose that a combination of high age, female sex, atrial fibrillation (above and beyond other somatic risk factors), low ADLs and depressive symptomatology should be

considered as significant (and partly modifiable) risk factors [8,9] for cognitive deterioration. Hence, we propose that a comprehensive treatment for early-stage AD calls for a multidisciplinary approach, including medical, (neuro)psychological and therapeutic professions.

## Supporting information

**S1 File.**
(DOCX)

## Acknowledgments

We thank the Austrian Alzheimer's Society for supporting this study. We wish to thank Ms. Cornelia Kirch for her valuable assistance in preparing the reference list.

## Author Contributions

**Conceptualization:** Liane Kaufmann, Elisabeth M. Weiss, Josef Marksteiner.

**Data curation:** Peter Dal-Bianco, Michaela Defrancesco, Gerhard Ransmayr, Reinhold Schmidt, Elisabeth Stögmann, Josef Marksteiner.

**Formal analysis:** Liane Kaufmann, Tilman Gruenbaum, Roman Janssen.

**Investigation:** Thomas Benke, Peter Dal-Bianco, Michaela Defrancesco, Gerhard Ransmayr, Reinhold Schmidt, Elisabeth Stögmann, Josef Marksteiner.

**Methodology:** Liane Kaufmann, Tilman Gruenbaum, Roman Janssen.

**Project administration:** Elisabeth M. Weiss, Thomas Benke, Peter Dal-Bianco, Michaela Defrancesco, Gerhard Ransmayr, Reinhold Schmidt, Elisabeth Stögmann, Josef Marksteiner.

**Writing – original draft:** Liane Kaufmann.

**Writing – review & editing:** Tilman Gruenbaum, Roman Janssen, Elisabeth M. Weiss, Thomas Benke, Peter Dal-Bianco, Michaela Defrancesco, Gerhard Ransmayr, Reinhold Schmidt, Elisabeth Stögmann, Josef Marksteiner.

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
