## [Decision Letter · Decision Letter 0]

19 Mar 2024

PONE-D-23-41598Predictive value of somatic and functional variables for cognitive deterioration for early-stage patients with Alzheimer’s Disease: Evidence from a prospective registry on dementia.PLOS ONE

Dear Dr. Kaufmann,

Thank you for submitting your manuscript to PLOS ONE. After careful consideration, we feel that it has merit but does not fully meet PLOS ONE’s publication criteria as it currently stands. Therefore, we invite you to submit a revised version of the manuscript that addresses the points raised during the review process.

Thank you for submitting your valuable work.

The reviews, which are insightful and interesting, pointed to some interesting points. The authors will notice the reviewers found merits in your study, but also raised important concerns.

By my own reading, tparticularly regarding its soundness, stats analysis, and the utilisation of graphs.

Also I have some additional suggestions. While these may appear lengthy, the intention is to align more closely with the recommendations while maintaining a high standard of scientific communication. As follows:

1) Please double check grammar throughout the text (e.g. there is incorrect comma usage, but what can be refined is avoiding jargon or overly complex sentences that may confuse readers unfamiliar with the topic);

2) Double check the references accordingly to the Journal’s standards (e.g. ref 42 is missing '10.1037//0882-7974.2.3.225', months are not required  but check it please, and only the first word of the title and proper nouns are capitalised);

3) The study encountered the challenge of a high dropout rate over the two-year period, decreasing from 500 participants at baseline to 169 at the final follow-up. While common in longitudinal studies, such attrition can bias the results and affect the study's power to detect changes over time. 

To mitigate this, the authors can indicate that following studies should implement measures to reduce participant dropout. This could involve maintaining more regular communication with participants, offering flexible scheduling options to accommodate their needs, and providing reinforcement for their involvement. Additionally, employing advanced statistical methods such as multiple imputation (i.e. if properly used) to handle missing data could aid in mitigating the biases introduced by participant attrition (Sterne et al., 2009). These steps not only demonstrate a commitment to maintaining participant engagement but also enhance the overall robustness and of the findings;

4) The study's findings from a specific cohort in Austria, may not be generalisable to broader populations due to cultural, genetic, and healthcare system differences. Expanding the research to broaden the research scope to encompass diverse populations across several countries would refine the external validity. This expansion would facilitate a more comprehensive understanding of Alzheimer's Disease (AD).

- Furthermore, including people from different educational and economic backgrounds is important. This helps us understand how AD affects various groups in society. By making research more inclusive, we can learn more about its characteristics. If possible, elaborate on past findings;

5) For instance, the study outlines the predictive value of demographic, somatic, and functional variables but may not sufficiently address the potential interaction between these factors. AD progression is influenced by a complex interplay of genetic, environmental, and lifestyle factors, and the additive or multiplicative effects of these variables could provide more insight into the disease’s progression. Plesae consider employing advanced stats such as structural equation to explore these interactions comprehensively (Livingston et al., 2020);

6) I'd highly suggest authors to cevaluate the model and predictive models, you can use cross-validation (e.g. train and test) or SVM as support for regression analyses. The 'MASS' package is excellent and user-friendly for various regression approaches. For SVM, you can try 'e1071' package. Additionally, for cross-validation, the 'rsample' package could be useful as it allows you to explore different approaches. These won't take more than few minutes.

- If the authors want an even more less time-consuming, you try JASP software;

- With multiple predictors and outcomes, there's a risk of false-positives. There is the lack of specification of (whether) corrections for multiple testing, such as the Bonferroni correction or FDR. Again, JASP is a very 'catchy' for this;

- Considering this, please also emphasise for other researchers that these adjustments to ensure the robustness of findings (Benjamini & Hochberg, 1995). These approaches could provide insights into the underlying mechanisms driving cognitive deterioration in AD;

7) The careful selection of covariates in regression models is essential for accurately estimating effects. However, the study lacks clarification regarding the reasoning behind choosing specific covariates over others. It is imperative to ensure a clear and theory-driven selection of covariates, supported by a pre-analysis plan. This approach helps avoid data dredging and promotes reproducibility (c.f. 10.1097/01.ede.0000056325.26393.17). To maintain the rigour of the findings, it is essential to employ methods such as multiple imputation or sensitivity analyses to assess the impact of missing data on the study's conclusions (c.f. 10.1037/1082-989X.7.2.147; 10.1002/9781119013563);

8) While caregiver reports are a practical method in AD research, enhancing them with objective measures or technology-assisted assessments could yield a more precise and holistic insight (c.f. 10.1177/0733464814543965) into the functional abilities and neuropsychiatric conditions of patients (c.f. 10.1093/geront/gnw250). For example, integrating wearable technology for continuous monitoring of physical activity could furnish objective data regarding activities of daily living (ADLs)- that is just an example;

9) The authors might suggest causal relationships between the predictor variables (e.g. such as atrial fibrillation and ADLs) and cognitive deterioration, even though there may not be convincing evidence of causality. 

- It's important to interpret these implications with caution, particularly in observational studies where other factors could be influencing the outcome;

10) The Discussion section might generalise the study's findings excessively, overlooking the limitations posed by the methodology, sample demographics, or data analysis.

- For example, suggesting that the identified predictors are universally applicable to all early-stage AD patients without acknowledging the cultural, genetic, and healthcare system diversity across different populations could restrict the relevance of the findings.

- However, by implementing the suggested techniques or approaches to enhance stats methods and refine findings, it's possible to explore these assertions without overextending the conclusions;

11) Also consider tidying the Tables and transferring some to the Sup. Material;

12) The utilisation of HQ-graphs in the Results section is important. The authors have remarkable and interesting data that could greatly benefit from exploration and visualisation for researchers and readers alike. 

- For regression analyses (Table 4), employing classic dot graphs with regression lines and scale-location graphs would be particularly insightful. Additionally, incorporating residual plots, especially for the predictors, such as partial regression graphs, could enhance understanding. These graph types can be found in packages like 'MASS' and 'ggfortify', extending beyond the capabilities of classic ggplot;

- For the correlation analysis (Table 5), the authors could either describe the correlations in plain text within the manuscript or utilise scatterplots with correlation coefficients to visually represent the relationships between variables;

- I'd highly suggest that the authors examine the autocorrelation function to assess the presence of seasonality. Additionally, for the LMM, it would be beneficial for the authors to present the main findings in plain text and complement them with representations over time. Polar graphs or line graphs depicting differences across time points could effectively illustrate these findings;

Please bear that these suggestions are intended to enhance the quality and presentation of your findings for researchers and readers. By implementing these recommendations, you can streamline your presentation by reducing the number of Tables and making your data more engaging. Given the potential of your study to attract a broad audience, it's essential to maintain rigorous approaches and ensure clear and engaging presentation, just as you have done with your study design and data collection.

Hope the authors find all (or most) of the suggestions useful and hope this helps somehow.

Wishing you success with the study

We look forward to receiving your revised manuscript.

Kind regards,

Thiago P. Fernandes, PhD

Academic Editor

PLOS ONE

2. In the online submission form, you indicated that [The data underlying the results presented in the study are available from Josef Marksteiner, MD (corresponding author, member of the PRODEM consortium).]. 

Reviewers' comments:

Reviewer's Responses to Questions

**Comments to the Author**

1. Is the manuscript technically sound, and do the data support the conclusions?

Reviewer #1: Yes

Reviewer #2: Yes

Reviewer #3: Yes

2. Has the statistical analysis been performed appropriately and rigorously? 

Reviewer #1: Yes

Reviewer #2: Yes

Reviewer #3: Yes

3. Have the authors made all data underlying the findings in their manuscript fully available?

Reviewer #1: Yes

Reviewer #2: Yes

Reviewer #3: Yes

4. Is the manuscript presented in an intelligible fashion and written in standard English?

Reviewer #1: Yes

Reviewer #2: Yes

Reviewer #3: Yes

5. Review Comments to the Author

Reviewer #1: Thank you for the opportunity to review this manuscript. The article was written well and covered a range of literature pertinent to the topic. The findings were also highly interesting and worthwhile for the academic community. I only have a few comments for the authors to consider.

Abstract

Lines 41-42 show an incomplete sentence.

Lines47-50 show repetition in reporting of results.

Introduction

Lines 107 and 109 – two acronyms are used here without explaining their definitions.

Method

Consent – can you clarify the consenting procedures used here? Were there any special considerations/accessibility requirements you had to meet for individuals diagnosed with cognitive decline (particularly as their dementia progressed)?

Data acquisition indicates that family caregivers were involved in the study but the participant eligibility criteria states ‘informal caregivers living with the patient’. Can you clarify this please?

Drop out analysis – could you quantify and report on the reasons for drop out instead of referring to other papers/studies?

General comments

‘Caregiver burden’ is increasingly being recognised as negative towards caregivers. Please consider more person-centred language such as ‘caregiving effects/challenges/load/impact’ as recommended in dementia language guidelines e.g., https://alzheimer.ca/sites/default/files/documents/Person-centred-language-guidelines_Alzheimer-Society.pdf

Reviewer #2: The authors have done a commendable job of presenting a study that assesses biological/somatic and functional variables associated with cognitive decline in early AD. The correlation between caregiver burden and patient-related measures in the early disease stages will be of particular interest to clinicians as they prepare their clinical interview/test battery. Finally, the generation of predictive variables that best explain cognitive decline over time is noteworthy.

The authors can consider fleshing out the discussion subsection ‘Predictive variables for cognitive deterioration’ to address the following:

- Contextualise how the fit of their model compares to others previously published that seek to determine the role of age/sex in contributing to cognitive decline risk. The authors could put forward rationale for the possible modest fit of their model. This comment is not meant to question the value of the current model.

- Provide a bit more explanation/speculation about why a) education was not a significant factors in their results/model and b) atrial fibrillation but not other somatic factors was a significant predictor of cognitive decline especially when the authors mention “The current scientific discourse on potential somatic risk factors, particularly those associated with cardiovascular health, remains inconclusive (lns 604-606). The authors have provided suitable ref(s) and general statements but a bit more explanation would be beneficial.

Minor points:

- Lns41-42: “variables for cognitive deterioration” (unclear sentence structure?)

- Ln75.. Section on dementia risk factors could include Livingston papers

- Table 1: caregiver sample size does not appear to add up

Reviewer #3: This is important topic and the paper is well organized and written. but couples of items should be addressed

1. remove CERAD, this is not the mesh key word

2. The current study seems very confusing and messy with multiple variables. Authors need to work more on the introduction to support 1) why caregiver burden is added to as one of the variables, it is well known from the previous studies, what are the lacking in this caregiver topic.

3. methods-results: all mixed together with measures, statistical analyses, and results, please write clearly to separate methods (measures and statistical analysis),

4. statistical analys: how did authors identify covariates, what covariates were controlled. 4. chi square test: authors need to address multiple comparison such as bonferronni correction. so many variables comparision will have type I error concerns of false positive results.

5. discussion need more work: what are not known (so many published papers already to show the relationships with psych, functional decline, and caregiver burden with AD). and address what parts of this study is adding to the existing results. Also, please address future intervention based on the results, and "so what" with the research findings

6. PLOS authors have the option to publish the peer review history of their article (what does this mean?). If published, this will include your full peer review and any attached files.

Reviewer #1: No

Reviewer #2: No

Reviewer #3: No

---

## [Author Response · Author response to Decision Letter 0]

11 Jun 2024

Dear Editor (Dear Dr. Fernandes), dear Reviewers, 

We would like to thank the Reviewers for their constructive and helpful comments. Upon revising the manuscript, we took great care to closely follow all comments raised by the Editor and the three Reviewers (see our response letter in the attachment).

As requested by the Editor, we now provide additional graphs and rerun several analyses by using alternative and – if requested – additional statistical methods (most of which we now report in the Supporting Material). Moreover, upon following the Editor’s comments, we now introduce a limitation section explicitly referring to the shortcomings of our study. In response to the Reviewer comments, we now (i) supplement the statistical methods section as well as the drop-out analyses (and provide more information on the reasons for drop-out); (ii) perform alpha corrections where appropriate; (iii) elaborate on the discussion (and how our findings add to the existing literature); and (iv) provide a rationale for our model fit. Overall, we believe that the quality of the manuscript did benefit significantly from the revisions. 

We confirm that neither the manuscript nor any parts of its content are currently under consideration or published in another journal. All authors have approved the revised version of the manuscript and agree with its resubmission to PLOS ONE.

We hope that you will find the revised manuscript appropriate for publication in PLOS ONE. 

Yours sincerely,

Liane Kaufmann, PhD (on behalf of all authors)

---

## [Editor Report · Decision Letter 1]

1 Jul 2024

Predictive value of somatic and functional variables for cognitive deterioration for early-stage patients with Alzheimer’s Disease: Evidence from a prospective registry on dementia.

PONE-D-23-41598R1

Dear Dr. Kaufmann,

We’re pleased to inform you that your manuscript has been judged scientifically suitable for publication and will be formally accepted for publication once it meets all outstanding technical requirements.

Kind regards,

Thiago P. Fernandes, PhD

Academic Editor

PLOS ONE

Additional Editor Comments (optional):

I have carefully scrutinised the rebuttal.

Thank you for your thoughtful and careful edits. The ms reads much better now, and I am confident that the concerns were properly addressed, even exceeding expectations in some aspects.

I'd suggest that the authors double-check the grammar (e.g. punctuation) and the references list again (e.g. ensuring consistency of studies not present in the list) during typesetting.
---

## [Editor Report · Acceptance letter]

17 Jul 2024

PONE-D-23-41598R1 

PLOS ONE

Dear Dr. Kaufmann, 

I'm pleased to inform you that your manuscript has been deemed suitable for publication in PLOS ONE. Congratulations! Your manuscript is now being handed over to our production team.

Kind regards, 

on behalf of

Dr. Thiago P. Fernandes 

Academic Editor

PLOS ONE